# Multimodal Cues Do Not Improve Predator Recognition in Green Toad Tadpoles

**DOI:** 10.3390/ani12192603

**Published:** 2022-09-28

**Authors:** Andrea Gazzola, Bianca Guadin, Alessandro Balestrieri, Daniele Pellitteri-Rosa

**Affiliations:** 1Department of Earth and Environmental Sciences, University of Pavia, 27100 Pavia, Italy; 2Department of Environmental Sciences and Policy, University of Milan, 20133 Milan, Italy

**Keywords:** amphibians, anti-predatory behaviour, chemical cues, visual cues, tadpoles

## Abstract

**Simple Summary:**

Tadpoles are known to use their sense of smell to detect the presence of predators, but some studies showed their reliance on vision during social interaction, suggesting that vision might have a role in predatory contexts as well. Here, we investigated how chemical or visual cues of a native predator, or a combination of both, influence the defensive behaviour of green toad tadpoles. We expected tadpoles to reduce their activity when exposed to chemical cues and avoid the area of the experimental arena near to the caged predator when exposed to the visual ones. With both cues, we expected tadpoles to show both responses and with greater intensity. Our results indicate that visual cues alone do not elicit any apparent defensive response, suggesting that tadpoles mainly rely on chemical cues to assess predation risk.

**Abstract:**

The anti-predator behaviour of green toad (*Bufotes balearicus*) tadpoles was investigated by exposing them to only the visual or chemical cues, or a combination of both, of a native predator, southern hawker *Aeshna cyanea*. We collected green toad egg strings in the field and tadpoles did not receive any predatory stimulus before the onset of the experiment. To manipulate chemical and visual cues independently, dragonfly larvae were caged inside a transparent plastic container, while chemical cues (odour of tadpole-fed dragonfly larvae) were injected into the surrounding arena. An empty container and water were used, respectively, as controls. The behaviour of individually tested tadpoles was videorecorded for 40 min, of which 20 were before their exposure to stimuli. Five second-distance frames were compared to assess both tadpole activity and position within the arena with respect to the visual stimulus. The tadpole level of activity strongly decreased after exposure to either chemical cues alone or in combination with visual cues, while visual cues alone apparently did not elicit any defensive response. The position of tadpoles inside the arena was not affected by visual cues, suggesting that green toad tadpoles mainly rely on olfactory cues to assess the level of predation risk.

## 1. Introduction

Predation is a main selective pressure for many species which strongly affects prey defensive responses aimed at avoiding detrimental consequences for fitness or, at worst, death [1]. In this context, the prey’s ability to perceive fluctuations in the level of predation risk lays the basis on which the nonlethal effects caused by predator occurrence are transmitted to both prey and predator populations [2]. From the prey perspective, a first step of vital importance is to gain information about the current local level of predation risk (“threat-sensitive assessment of predation risk” hypothesis; [3,4]) through one or more sensory modalities [5], depending upon prey specific adaptations and environmental conditions.

For example, predatory pressure from bats is a key factor shaping the ability of insects to hear specific sound frequencies, or their sensitivity to them [6]. Accordingly, prey populations that, over generations, have not been in contact with bat predators have weakened their sensitivity to bat echolocations [7]. Rodents, which are prey for a wide range of carnivores, have evolved the capacity to sense predator odours with very diverse chemical structures, which are identified by different components of their olfactory system [8], which is a fundamental adaptation to enhance their chance of survival.

In anuran larvae, the main sensory pathway employed to assess predator presence is olfaction [9,10,11]. Many studies have shown the capacity of larval anurans to respond to chemical cues produced by the predator alone (kairomones) or borne from predation on conspecific or heterospecific prey (alarm signals) (reviewed in [12,13]), but other sensory modalities may play a role in risk detection, including audition, mechanoreception (by means of the lateral line), and vision [14].

Here, we adopt the generally accepted convention of using the term of theatrical origin “cue” to indicate information that is delivered unintentionally by the sender (e.g., predator) and is profitably used only by the receiver [15]. The effects of visual cues have been explored less extensively than chemical ones and are generally considered of minor importance for amphibian larvae, which seem to be near-sighted [16,17,18], i.e., incapable of fine-scale discrimination, also at a very short distance.

However, in non-predatory contexts, tadpoles have been observed to rely on visual cues when correcting swimming activity in arrangement with other individuals [19]; adjusting growth and development in relation to the simulated presence of conspecifics (produced by mirrors; [20]) or choosing among groups of individuals of different numerousness [21].

The use of visual information during social interaction leaves room for its possible role in predatory contexts. The capacity of animals to rely on different types of sensory cues is expected to enhance their chance of escaping predation. The use of a multimodal cue is especially useful when habitat conditions prevent the efficiency of one sensory pathway, making the use of other sensory modalities of paramount importance to assess predator presence. In aquatic environments, olfactory cues may allow for the detection of the presence of predators, while visual cues may provide reliable information on their position and distance, and hence on the level of predation risk [22].

To investigate the relative importance of sensory pathways on the assessment of predation risk by anuran larvae, we explored the defensive behaviour of green toad tadpoles (*Bufotes balearicus*) in response to larval dragonfly chemical cues, visual cues, or a combination of both. Based on two well-known behavioural responses that tadpoles display when exposed to predators, that is activity decrease and spatial avoidance [23,24], we expected green toad tadpoles to alter their behaviour whenever their sensory systems would detect an actual hint of predation risk. We hypothesised that, when relying on olfactory cues, tadpoles would reduce their level of activity, while predator visual cues would elicit both spatial avoidance and, secondarily, activity reduction. When exposed to both cues, we expected tadpoles to display both defensive responses, with increased intensity with respect to unimodal cues.

## 2. Materials and Methods

### 2.1. Animals Collection

In May 2020 six freshly laid green toad egg strings were collected from a network of canals flowing in an intensively cultivated area east of Milan (45°26′ N, 9°20′ E, Lombardy region, Northern Italy). Egg clutches were immediately carried to the laboratory and prepared for the experiments. After hatching, tadpoles were transferred in 18 different containers (15 L) to be raised in similar density conditions. The day before the onset of the experiment, 40 tadpoles were selected from each string (29–33 Gosner stage) and inserted into a 150 L plastic container filled with 60 L of dechlorinated water. Tadpoles were visually selected to obtain individuals of approximatively similar size. Twenty late instar southern hawker dragonfly larvae (*Aeshna cyanea*), which are widespread native predators of anuran tadpoles, were collected from an artificial pond located inside the protected natural area “Bosco del Vignolo” (45°13′ N, 8°56′ E, Lombardy region, Northern Italy) using dip-nets. In the laboratory, they were kept individually in 0.8 L plastic containers, filled with 0.5 L of dechlorinated water, and containing a mesh fragment (4 cm × 4 cm).

### 2.2. Experimental Design

The experiment consisted of evaluating the defensive response of green toad tadpoles elicited by olfactory and visual cues of dragonfly larvae. We combined two factors with two levels each, that is presence or absence of olfactory cues and presence or absence of visual cues, in a 2 × 2 full design. We conducted trials of 45 min in total, testing one tadpole per arena. Each tadpole was randomly assigned to a specific combination of sensory cues (visual cue alone, olfactory cue alone, or a combination of both cues) or a control treatment (no cues). The experimental arena (Figure 1) consisted of a white opaque plastic tub (30 cm × 6 cm × 22.5 cm) filled with 1 L of dechlorinated water; a rectangular transparent plastic container (5.50 cm × 5 cm × 4.9 cm), in which an individual predator could be present or not (visual stimulus), was positioned in contact with one of the short inner sides of the arena. The opposite side was left empty. The “predator” container was initially covered with a green plastic sheet, which formed a closed polygonal barrier around the box and prevented tadpoles from seeing its contents. The barrier was connected to a wire, which allowed it to be easily removed without disturbing the tadpole. The focal tadpole was put in the center of the arena surrounded by a circular net (6 cm diameter) for 5 min to allow for acclimatation. After this period, the videorecording started and the net was removed, allowing the tadpole to move freely within the arena. During this pre-stimulus period, no cue (visual or chemical) was available for tadpoles to be detected. After 20 min, the plastic barrier (covering the visual stimulus) was removed and, eventually, the predator’s cue was injected in the arena (olfactory stimulus); the post-stimulus videorecording lasted 20 min. For each behavioural trial, we recorded four different individual tadpoles by creating a grid of five arenas, which was positioned inside a large opaque plastic container (63 cm × 85 cm × 50 cm) to reduce external disturbances. All tests were performed indoors and video-recorded with a digital camera Canon Legria (Canon, Tokyo, Japan) hung up 1.2 m above the arenas. We tested 24 individuals for each of the four predator treatments (overall, *n* = 96 tadpoles).

Each of the four recording days, olfactory cues were obtained from five dragonfly larvae, randomly selected from a set of 12 individuals, which were fed with green toad tadpoles one hour (8:30 a.m.) before the onset of the experiment. Each predator was provided with two anuran larvae (200 mg), which were preyed on within a few minutes. Just before the start of each daily recording session, 50 mL of water were collected from each predator tub and pooled into a single container. The olfactory stimulus consisted of 8 mL of the mixed solution, which was gently injected by a 10 mL syringe inside the experimental arena (Figure 1). The concentration of the cue (1:125) was consistent with previous experiments [11,25,26]. After the collection of the cues, a complete water change was made for all predator containers. For visual stimuli, predators were used 2–3 times each, and assigned haphazardly to each recording session. At the end of the experiments all tadpoles and dragonflies were released at their site of capture.

### 2.3. Data Collection and Statistical Analysis

To assess tadpole behavioural responses, all videos were visually inspected by the same observer in a semi-blind condition (blind to the chemical treatment while inevitably not to the visual one). Both tadpole and predator activity were assessed by comparing consecutive frames at five second intervals (for a total of 46,080 observations), recording activity as a dichotomous variable, 0 (inactive) or 1 (active, i.e., showing frame to frame movements longer than tadpole body width).

To explore the effect of the visual stimulus, for each frame we also assessed tadpole position within the experimental arena with respect to the “predator” container. The experimental arena was divided by a median line, and, for each frame, tadpole position was filed as 0 when inside the half part of the arena with the “predator” or 1 when inside the opposite half (i.e., far from the visual stimulus). Response variables for statistical analysis were obtained by calculating the proportion of active-tadpole observations and the proportion of far-from-predator observations. Both response variables ranged within the [0,1] interval. We ran two generalized linear mixed models (GLMMs); both models included treatment and the pre-stimulus covariate (activity or position) as fixed effects, and the trial as a random effect. For activity, we used a beta distribution of the residuals and logit as a link function, while a Gaussian distribution was used for the position of the tadpole in the experimental arena.

As the beta distribution requires data with observations in the open range (0, 1) [27], we transformed both the response variable and covariate (position before stimulus injection). Considering the interval [a, b], we rescaled to (0, 1) using the formula $\frac{x-a}{b-a}$ and, subsequently, applied the formula
(1)$$y=\frac{x\left(N-1\right)+0.5}{N},$$
where *N* is the total number of observations and *x* the original response variable. For tadpole position, we also ran a model on a long format dataset, with injection (factor with pre- and post-stimulus as levels), treatment, and their interaction as fixed effects, and the trial as a random effect. This model allowed the estimation of the difference between tadpole mean position and the reference level of no choice (=0.5), and also the exploration of variations after exposure to the cues.

Finally, we explored the potential effect of predator activity on tadpole activity, using the half set of data implying predator presence (i.e., visual and visual + olfactory treatments). We applied a linear model with tadpole activity as the response variable and predator activity as the continuous predictor. Statistical analyses were performed using R (version 3.6.0) and package glmmTMB to run GLMMs [28]. Package emmeans [29] was used to extract estimated means and dispersion measures from the models.

## 3. Results

The model showed that tadpole post-stimulus activity was affected by both pre-stimulus activity (χ^2^ = 6.68, df = 1, *p* = 0.01) and treatment (χ^2^ = 98.23, df = 3, *p* < 0.0001). Olfactory cues, both alone and in combination with visual cues, induced a significant decrease in tadpole activity in comparison to controls and visual cues (Figure 2 and Figure 3).

Tadpoles showed a general tendency to spend more time in the half of the arena where the transparent container was present (all means, for all treatments both pre- and post-stimulus, were <0.5); except for the pre-stimulus control and olfactory + visual treatments, these differences were significant (highest *p* = 0.03). This tendency was, in general, further increased by cues, with the post-stimulus mean of the multimodal stimulus treatment differing significantly from the corresponding pre-stimulus (pre-post estimated difference = 0.09, SE = 0.04, df = 182, t-ratio = 2.18, *p* = 0.02).

Tadpole position was not affected by either treatment (χ^2^ = 0.41, df = 1, *p* = 0.51) or pre-stimulus distance (χ^2^ = 1.60, df = 3, *p* = 0.65); actually, the mean values of all treatments were similar to those recorded for the control group (Figure 4). Finally, the tadpole level of activity was not affected by predator activity, as the slopes of both visual and multimodal treatments did not significantly differ from zero (Figure 5; slope = 0.04, SE = 0.26, df = 42, 95% CI [−0.49, 0.57] and slope = −0.29, SE = 0.22, df = 42, 95% CI [−0.72, 0.15], respectively), and from each other (estimated difference = 0.33, SE = 0.34, df = 42, t-ratio = 0.97, *p* = 0.33).

## 4. Discussion

As expected [21,30], green toad tadpoles responded to olfactory cues of native odonate predators by strongly decreasing their activity level, a behavioural response which has been recorded in several anuran larvae [31]. Oppositely, both the results of the visual treatment and the tadpole response to the presence of the predator, as assessed by analysing tadpole distance from the “predator” container and comparing the relative activity of predator and prey, suggest that tadpoles mainly rely on olfactory stimuli to estimate predation threat and exhibit a clear defensive response.

Our results agree with most previous studies, which did not record anything more than weak responses to visual cues of potential predators by tadpoles of several anuran species (*Rana lessonae* and *Rana esculenta*, [32]; *Bufo boreas*, [33]; *Mannophryne trinitatis*, [34], 2006; *Rana pipiens*, [35]; *Sphaerotheca breviceps* and *Bufo melanostictus*, [36]; *Allobates femoralis*, [37]).

A remarkable exception was reported by Hettyey et al. [14] who recorded lowered activity in *Rana temporaria* tadpoles exposed to visual cues of two predators, dragonfly larvae and three-spined sticklebacks, *Gasterosteus aculeatus,* suggesting that either the larval stage (tadpole visual ability improves during development [38]), or laboratory conditions (size of the testing arena) may affect tadpole responses to visual cues.

Szabo et al. [37] investigated the response of neotropical poison frog *Allobates femoralis* exposed to visual and chemical cues of either a larval odonate predator or heterospecific predatory *Dendrobates tinctorius* tadpoles. Although the response was weak, differing from our results, tadpoles avoided the glass cylinder containing the odonate predator when also exposed to its odour, while their behaviour was not affected by the presence of the heterospecific predator. These contrasting results suggest that, as both the predator’s morphological traits, such as shape, colour, or size, and numerosity (solitary vs. social predators) may affect the visual perception of potential threats by tadpoles, the role of multimodal cues should be investigated using a wide range of predators.

Finally, many anuran species have been reported to learn predator dangerousness by experiencing, at the same time, the predator odour and alarm chemical cues released by injured conspecifics [39,40]. As our tadpoles never had the opportunity to experience any predator species, we could not assess the role played by learning in the identification of a potential source of risk by visual cues.

The investigation of the effects of different predatory sensory cues on prey risk perception implies a main experimental issue that needs to be carefully addressed, that is the experimental setting must allow for the effective distinguishment of the effects of each type of cue. This issue is not as easy to fix as it may seem at first glance. When using both olfactory and visual cues, the usual approach is the use of different containers, e.g., transparent containers will expose the visual cues while preventing chemical cues from diluting, while cheesecloth nets will do the opposite [41].

Notwithstanding, other types of cues may interfere with the experimental design, for example, acoustic and hydraulic cues cannot be completely restrained by any of these containers or, at worst, they may be transmitted with a different intensity, thus generating a confounding factor which will undermine the results of the experiment.

This issue was partially overcome by Stauffer and Semlitsch [32], whose experimental setting included three separated containers inside the same experimental aquarium, one for visual cues, one for chemical cues, and one for tactile cues. In this setting, and by alternatively blocking either visual or chemical cues, the disturbing effect of potential confounding factors was partially averted; in fact, using the authors’ protocol, undesired cues should have the same intensity in all arenas, except for possible random fluctuations.

Another useful approach, which may improve an experimental design aiming to detect the effect of visual cues, is to plan a further control consisting of a transparent tub containing tadpoles of the focal species as a neutral visual stimulus [18]. In this last case, arguably controlling for movements produced by different visual treatments, the experiment may allow for the discrimination between the actual recognition of a potential predator species and a generic visual stimulus. Notwithstanding, the tendency of anuran larvae to aggregate with conspecifics [21] may bias the effectiveness of controls.

Despite the fact that the experimental setting we adopted had some weaknesses, e.g., a single transparent container inside the experimental arena (instead of two, the second acting as further control for visual cues), or the lack of control for mechanical and acoustic cues, differently from other studies, we also recorded tadpole behaviour during the pre-stimulus phase, which was intended to improve the reliability of behavioural assessments after tadpole exposure to stimuli [42] by controlling for inter- individual variation in the basal level of activity.

Besides including different control treatments, a useful strategy may also consider transparent containers differing in shape or materials, or different contrasts between the visual stimulus and the background of the container; although they require more preparatory work to settle the experiment, these expedients may effectively improve future studies aiming to unravel the relative importance played by sensory cues in assessing predation threat.

## 5. Conclusions

A major question about visual cues is raised by tadpole responses to the sight of conspecifics, which has been reported to affect aggregation [21], growth and the duration of the larval period [43], and activity levels [44]. This capability, which can be expected to differ between social and solitary species [21], may imply that the visual perception of conspecifics, as either partners or potential competitors, is a selective pressure stronger than predation threat, which actually may materialize through a plethora of carnivorous species differing in morphology and behaviour, depending on local environmental conditions and communities. In these terms, the association of chemical cues may be a cost-effective strategy to fine-tune defensive responses, which needs to be further investigated.

## Figures and Tables

**Figure 1 animals-12-02603-f001:**
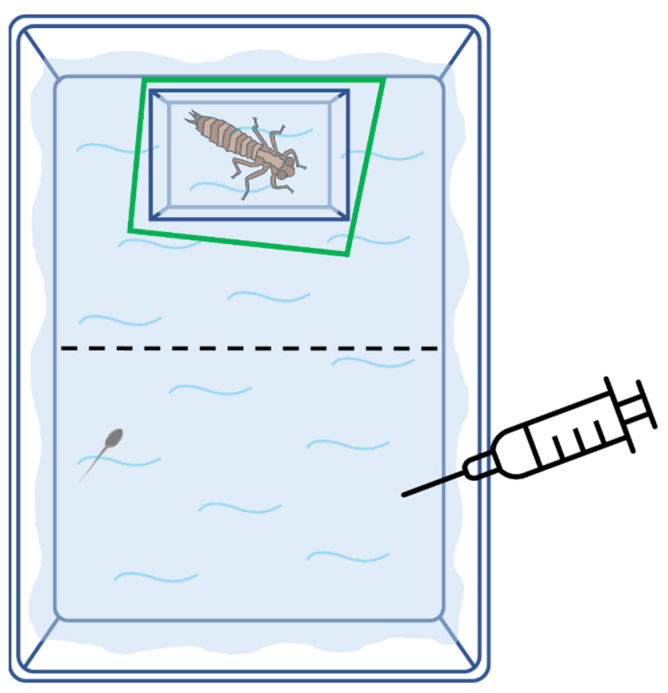
Experimental arena. The predator is located inside a transparent container (top), while the tadpole can freely move in both halves of the arena. After 20 min (pre-stimulus period) from the beginning of the trial, the green polygonal was removed and 8 mL of the olfactory cue (water or predator cue) was added into the arena using a syringe.

**Figure 2 animals-12-02603-f002:**
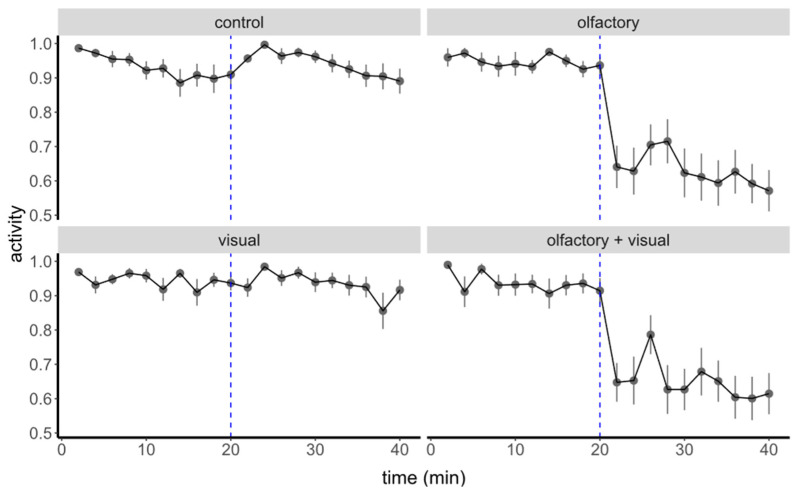
Mean proportion of active tadpoles during the 20 min pre- and post-stimulus. Each point represents a mean obtained from 24 individuals within a 2-min interval (overall, *n* = 96). Blue dashed lines indicate the moment of cue presentation, at the twentieth minute from the start of the experimental trial.

**Figure 3 animals-12-02603-f003:**
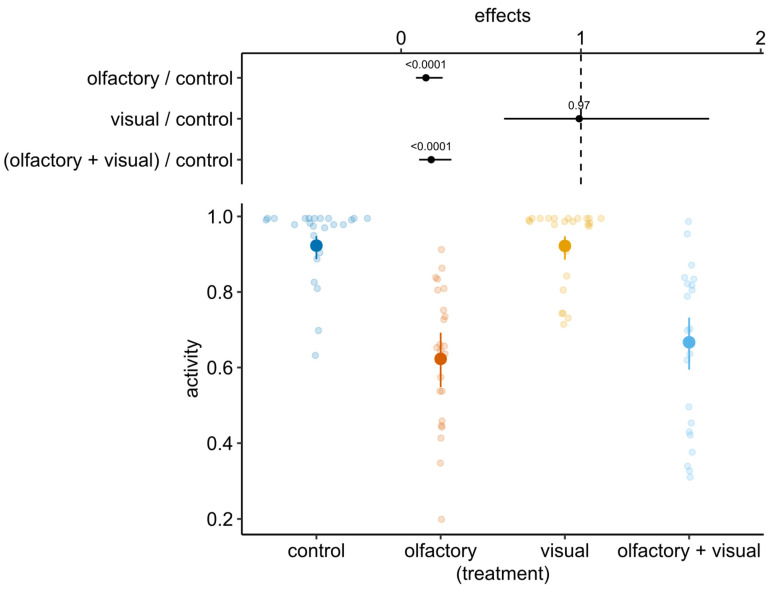
Estimated means of tadpole post-stimulus activity (larger coloured points) and relative 95% CI (vertical bars) from the beta mixed model. The top of the plot shows the estimated effects (ratio) as comparison with control treatment (absence of both predator visual and olfactory cues); estimates not overlapping with the vertical dashed line (i.e., ratio = 1) indicate significant difference in comparison with control treatment.

**Figure 4 animals-12-02603-f004:**
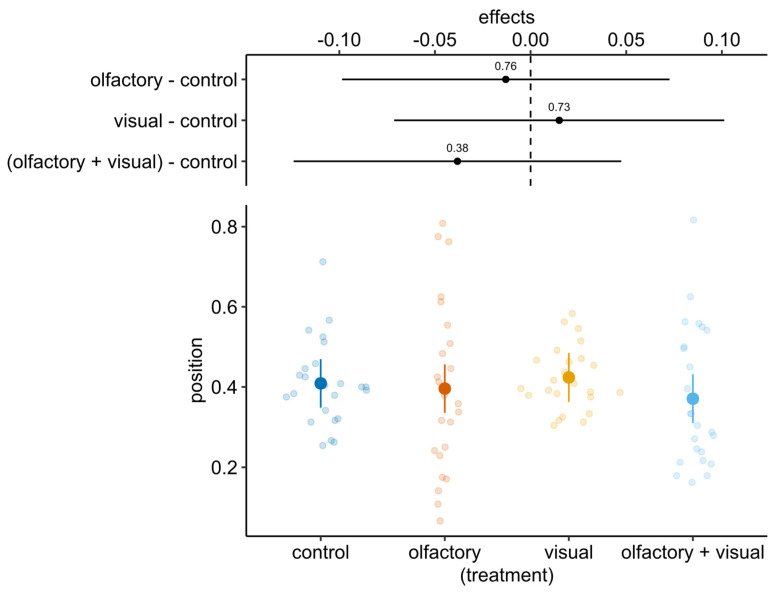
Estimated means of tadpole post-stimulus position (larger coloured points) and relative 95% CI (vertical bars) from the mixed model. Coloured dots indicate estimated individual means, where values below 0.5 indicate a preference for the portion of the arena which contains the transparent container. The top of the plot shows the estimated effects (difference) in comparison with control treatment (absence of both predator visual and olfactory cues); estimates not overlapping with the vertical dashed line (i.e., difference = 0) indicate significant difference in comparison to control treatment.

**Figure 5 animals-12-02603-f005:**
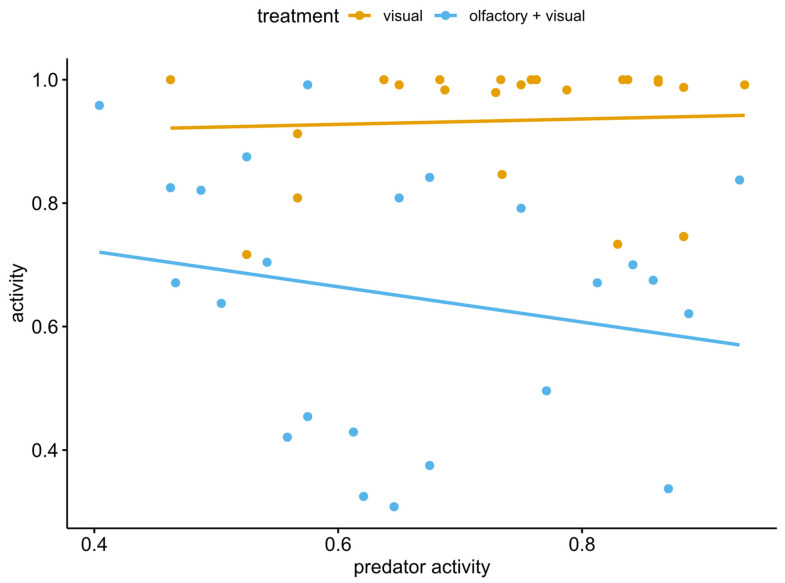
Relationship between tadpole post-stimulus activity and predator activity for both visual and olfactory + visual cue treatments.

## Data Availability

Data are available on request from the corresponding author.

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
