# Peer review of "Multimodal Cues Do Not Improve Predator Recognition in Green Toad Tadpoles"

_animals, 2022, doi:10.3390/ani12192603_

Round 1

Reviewer 1 Report

Comments of Manuscript “Multimodal Cues Do Not Improve Predator Recognition in 2Green Toad Tadpoles”. Manuscript ID-animals-1928049

Summary

In this manuscript authors evaluate the antipredator behavior in green toad tadpoles studying whether tadpoles can detect risk via visual, chemical cues, or both. They performed an experimental design by exposing tadpoles to visual and chemical cues or the combination of both and measuring level of activity and position of tadpoles in the experimental arena. In general, the manuscript is well presented but I have some concerns about discussion and conclusions. Language is clear and unambiguous permitting the reader to fluently follow the text. Included in the manuscript there is much information about how prey respond to predators by modifying traits, and many of these examples belonging to anuran tadpole studies. More of the studies have explored how predator risk affects activity and there is a lot of work showing a general pattern with tadpoles drastically reducing their activity when predation risk is present. Most of these studies mainly focus on the effect of chemical cues (alarm-conspecific cues or kairomones from predators), however less information is available in the literature about the effect of visual cues from predators to tadpoles. Then, it represents a good topic to be investigated. The study is well proposed; the experimental protocol seems to be adequate to respond the questions proposed and in general, the results are adequately described. The manuscript contributes to the general knowledge of how prey perceive risk in a context of non-lethal predation risk scenario; however, I find problems when the authors discuss their results and emphasize on methodological problems. I suggest to improve the discussion focusing in their results, more than take an extent explanation on the weakness of the methodological protocols. There is a lot of work published, for example, by themselves (in the same species studied in this work) and from the literature to compare with the results and propose hypotheses about the non-visual effect of the predator.

General comments

The manuscript is well structured and abundant of specific literature reviewing the topic with recent and relevant publications. The experimental design is in general, appropriate to respond the objectives proposed. Results follow the methods and are adequately described. However, I noted that discussion is mainly focused in the weakness of experimental studies such this study, and emphasizing of the many interferences and difficulties to isolate the main variables to be measured when studying prey behaviors; then, I perceive the authors made an excessive explanation on this issue, and the problems they found when trying to establishment adequate protocols. My suggestion is to focus and discuss the specific results of this study respect to the (abundant) literature available of tadpole responses to predation risk.

Specific comments

Statistical analysis is well explained and developed.

Figures are clearly presented, and adequate to explain the observed results.

I have two main concerns, discussion and conclusions. As I mentioned above, discussion section needs to be improved focusing on their own results and comparing what is observed in literature, with mention of the weakness of these kinds of experimental protocols but not being the main concern. But my suggestion is to emphasize in the observed results and whether new hypothesis arise. For example, author have worked with this and another species on the behavior of tadpoles analyzing effects of group living in the defensive behavior. Could this type of behavior be affecting visual detection in this species based on previous studies? Or, Do the authors considered external factors as complexity of the environments?

Finally, I cannot see in the conclusions relationship with the results show in the study. The relevance of aggregation in tadpoles throughout the study was barely mentioned, then I suggest focus on the actually observed results, their relevance and weakness, and future research addressing new questions emerging from this study.

Author Response

General comments

The manuscript is well structured and abundant of specific literature reviewing the topic with recent and relevant publications. The experimental design is in general, appropriate to respond the objectives proposed. Results follow the methods and are adequately described. However, I noted that discussion is mainly focused in the weakness of experimental studies such this study, and emphasizing of the many interferences and difficulties to isolate the main variables to be measured when studying prey behaviors; then, I perceive the authors made an excessive explanation on this issue, and the problems they found when trying to establishment adequate protocols. My suggestion is to focus and discuss the specific results of this study respect to the (abundant) literature available of tadpole responses to predation risk

AR. We thank the reviewer for the general positive comment. We agree with the reviewer and have integrated the discussion as suggested. However, we preferred to retain the part of the discussion on the experimental design.

Specific comments

Statistical analysis is well explained and developed.

Figures are clearly presented, and adequate to explain the observed results.

I have two main concerns, discussion and conclusions. As I mentioned above, discussion section needs to be improved focusing on their own results and comparing what is observed in literature, with mention of the weakness of these kinds of experimental protocols but not being the main concern. But my suggestion is to emphasize in the observed results and whether new hypothesis arise. For example, author have worked with this and another species on the behavior of tadpoles analyzing effects of group living in the defensive behavior. Could this type of behavior be affecting visual detection in this species based on previous studies? Or, Do the authors considered external factors as complexity of the environments?

Finally, I cannot see in the conclusions relationship with the results show in the study. The relevance of aggregation in tadpoles throughout the study was barely mentioned, then I suggest focus on the actually observed results, their relevance and weakness, and future research addressing new questions emerging from this study

AR. The discussion has been integrated with further comparisons between our and previous results. The relevance and weakness of our study have been highlighted in the discussion section. In the conclusions we suggest the need for investigating both the role of chemical cues, namely if they may be more effective for detecting predators, and potential variation in the role of visual cues in relation to the social behaviour of prey.

Reviewer 2 Report

This manuscript reports on a small experiment investigating behavioural responses of tadpoles to predator cues in a simple artificial setting. The main result – activity reduction in the presence of olfactory cues – matches expectations based on many studies of many anuran species. Effects of visual cues were not detected.

Experimental design and results are clearly presented (for a few minor issues, see below). Results are straightforward for olfactory, but remain inconclusive with regard to visual cues. A large part of the discussion deals with methodical issues, in particular the presentation of visual cues, and separating effects of different cues.

Some points that are not touched, but may be of interest for the reader: How often do tadpoles of green toads co-occur with Aeshna larvae? The experimental literature on tadpole-predator interactions seems strongly biased towards dragonfly larvae, probably because they can be easily handled in the lab. I guess for green toad tadpoles other, more mobile predators may be more relevant in the field (diving beetles, water bugs; snakes, birds). For detection of these predators, visual and mechanic cues may be more important than in the case of dragonfly larvae (which are sit-and-wait predators often associated with vegetation or other structures in the field).

I have severe misgivings about the title. "Multimodal cues do not improve" seems wrong for two reasons: Only two cues from a single predator species were tested, therefore no generalisation on multimodal cues is warranted; and it is unclear what is the reference of "improve".

line 93: "40 tadpoles ... from each string"; were these tadpoles evenly distributed over the four treatments, was sibship used as factor in the statistical analyses? It might be of interest if a part of the large variation in activity levels was related to sibship.

line 102 (and elsewhere): I am not sure whether "defensive response" is the best word here; these behavioural responses seem to be evasive rather than defensive.

line 151: Is "body depth" the correct term here (I would assume it refers to the vertical extension of the body, which can hardly be assessed from above)?

Author Response

Some points that are not touched, but may be of interest for the reader: How often do tadpoles of green toads co-occur with Aeshna larvae? The experimental literature on tadpole-predator interactions seems strongly biased towards dragonfly larvae, probably because they can be easily handled in the lab. I guess for green toad tadpoles other, more mobile predators may be more relevant in the field (diving beetles, water bugs; snakes, birds). For detection of these predators, visual and mechanic cues may be more important than in the case of dragonfly larvae (which are sit-and-wait predators often associated with vegetation or other structures in the field).

AR. The discussion has been integrated with further comparisons between our and previous results.

I have severe misgivings about the title. "Multimodal cues do not improve" seems wrong for two reasons: Only two cues from a single predator species were tested, therefore no generalisation on multimodal cues is warranted; and it is unclear what is the reference of "improve".

AR. We disagree with the reviewer on this point. As with other similar studies, the aim of our chosen title was to give readers an immediate idea of the type of research that was done in this paper, without trying to express a generalized view of the effect of multimodal cues. First, however, as for the term “multimodal”, the experiment includes a condition in which two types of cue (chemical and visual) were present at the same time and, as reported in other studies, the use of the term “multimodal” is appropriate (meaning more than one type of cue). E.g. in human communication, “perception of speech sounds is modulated by observation of the gesture”, and it is considered multimodal. Second, we understand the limitations given by using only one predator species, and to clear this point we added a phrase in the Discussion, in line 251 (“the role of multimodal cues should be investigated using a wide range of predators”). Third, the general effect of using more than one cue is that prey may integrate the information and express a stronger response than that elicited by a unimodal cue. As such, this enhanced response is expected, and the expression “not improve” just indicates an unexpected result of what was logically predicted.

line 93: "40 tadpoles ... from each string"; were these tadpoles evenly distributed over the four treatments, was sibship used as factor in the statistical analyses? It might be of interest if a part of the large variation in activity levels was related to sibship.

AR. As reported “40 tadpoles were selected from each string (29-33 Gosner stage) and inserted in a 150 L plastic container”, tadpoles from different strings were mixed in single container and, consequently, the strain was not considered in the analysis.

line 102 (and elsewhere): I am not sure whether "defensive response" is the best word here; these behavioural responses seem to be evasive rather than defensive.

AR. “defensive response” can actually take different forms, and this term is frequently used in behavioural and ecological studies to describe many types of behaviour. For example, tadpoles (as many prey species) can express diverse anti-predatory behaviours like activity reduction, spatial avoidance or hiding, which are unanimously referred to as “defensive responses”.

line 151: Is "body depth" the correct term here (I would assume it refers to the vertical extension of the body, which can hardly be assessed from above)?

AR. We changed the term with “body width”.